# Task Plan verbalizations with causal justifications

**Gerard Canal,**[1] **Senka Krivić,**[1] **Paul Luff,**[2] **Andrew Coles**[1]

[1] Department of Informatics, King's College London
[2] King's Business School, King's College London
{name.surname}@kcl.ac.uk

## Abstract

To increase user trust in planning algorithms, users must be able to understand the output of the planner while getting some notion of the underlying reasons for the action selection. The output of task planners have not been traditionally user-friendly, often consisting of sequences of parametrised actions or task networks, which may not be practical for lay and non-expert users who may find it easier to read natural language descriptions. In this paper, we propose PlanVerb, a domain and planner-independent method for the verbalization of task plans based on semantic tagging of the actions and predicates. Our method can generate natural language descriptions of plans including explanations of causality between actions. The verbalized plans can be summarized by compressing the actions that act on the same parameters. We further extend the concept of *verbalization space*, previously applied to robot navigation, and apply it to planning to generate different kinds of plan descriptions depending on the needs or preferences of the user. Our method can deal with PDDL and RDDL domains, provided that they are tagged accordingly. We evaluate our results with a user survey that shows that users can read our automatically generated plan descriptions, and are able to successfully answer questions about the plan. We believe methods like the one we propose can be used to foster trust in planning algorithms in a wide range of domains and applications.

## Introduction

The plans produced by a task planner may not be easy to understand by lay users and people not familiar with planning. This plan output, usually written as a sequence of parametrised actions, does not integrate enough information for users who are not experts in the domain to understand it and the possible reasons for each action in the plan.

These users may be more familiar with natural language descriptions of the plans, narrated as a sequence of sentences describing the actions and involving the parameters. Furthermore, this narration of the plan can include causality information to link the actions together, making more explicit why a specific action was taken. We believe this would make it even easier for those users to understand the plan, possibly increasing their trust in the planner. Additionally, this may also enable the planning systems to narrate the plan themselves, fostering interaction with the user. A clear example of this would be that of a robot acting in a human-inhabited environment and explaining its plans to the users around.

In this paper, we present PlanVerb, a domain-independent method to verbalize task plans for planners based on PDDL (Fox and Long 2003) and RDDL (Sanner 2010). For this, we first propose semantic tagging for planning domains that specify the building blocks of the verbalized sentences (verb, subject, and complements). The tags are used by PlanVerb to generate the sentences, but may also be useful for readers of the domain to get a quick idea of what each action represents. We also present an action compression method with the intent of summarising plans by joining together actions that act on the same parameters. An example of this are compressions of navigation actions which go through an intermediate point. Finally, we propose an extension to the verbalization space parameters from (Rosenthal, Selvaraj, and Veloso 2016), previously used to narrate robot navigation routes. These parameters represent user preferences on the narration, and allow the generation of verbalizations at different levels of detail including only certain actions or objects, and with more or fewer causality explanations. A user evaluation with 42 participants has demonstrated that the proposed approach generates understandable plan verbalizations.

## Related work

This work on task plan verbalizations extends the work by Rosenthal, Selvaraj, and Veloso (2016), where verbalization is applied to the narration of mobile navigation routes. In that work, the authors introduce a verbalization space that covers the variability in utterances that can be used to describe the route to different users. The route and map of the robot are used to instantiate sentences that narrate the robot experience. Then they performed a user study in (Perera et al. 2016) where they analysed the kinds of questions that the users can request to the robots to obtain the desired explanations, to then learn a mapping between user queries and verbalization space parameters. This approach was further adapted in (Zhu et al. 2017) to narrate manipulation tasks along with navigation, including PDDL actions. We have extended this notion of verbalization and verbalization spaces and applied to task plans in a domain-independent fashion and integrating causality information to explain the relations between actions, with applications not restricted to robotics. Furthermore, we don't need pre-writing sentence

templates, but only tagging the syntactic elements of the actions in the domain.

Verbal communication of plans has been deemed necessary in robotic scenarios involving humans. Fiore, Clodic, and Alami (2016) verbalize the actions in the plan for the user, explaining which actions be executed and in what order. Canal, Alenyà, and Torras (2019) communicate the next action in the plan when the "inform" action, which is part of the plan, is executed. Both works provide domain-dependent verbalization of the plans, probably written specifically for the task to be performed. In (Singh et al. 2020), robot teams verbalize explanations of their actions and intentions to increase human understanding. The plan is verbalized by partitioning it based on the informativeness of the actions. The utterances come from predefined templates of possible words and phrases. Similarly, Nikolaidis et al. (2018) explore how utterances improve Human-Robot Collaboration with a robot that issues commands to users and explains why it is doing some actions. The proposed formalism combines the verbal communications and the robot actions optimally to improve task performance. Neither of these works make causal relation between actions explicit, which may help the users understand the reasoning behind the actions.

State verbalization was performed in (Moon et al. 2019), where language descriptions of scene graphs are verbalized and used for scene understanding to describe the states while executing the plan, although these descriptions are not yet linked with the planning domain or planner.

Hayes and Shah (2017) explain robot control policies, verbalizing learned action conditions queried by the user. Similar to our domain description tagging, they add function decorators in the code to be able to verbalize the actions performed by the robot. Sridharan and Meadows (2019) present a theory of explanations for Human-Robot Collaboration. With it, they represent, reason, and learn knowledge to generate explanations, an explanation categorisation, and an explanation construction method. The defined characteristic axes can be seen as an equivalent of the verbalization space. Causal chains have been used to provide explanations in Seegebarth et al. (2012), where plans are represented in first-order logic with explanations being proofs based on causal links. Madumal et al. (2020) also use causal chains to generate explanations for RL agents using decision trees.

We summarise the plans by compressing some of the actions appearing in it. This is similar to work performed on Macro-Operators (Botea et al. 2005; Coles, Fox, and Smith 2007), where a set of ground actions are joined to form a macro-action. Similarly, we join sets of actions operating in intermediate parameters to verbalize them together. Other summarisation approaches, such as (Myers 2006), perform summarisation by only describing features based on semantic concepts, while we compress redundant parts of the plan to show it as a whole.

## Semantic domain information tagging

In order to generate sound sentences that represent each of the actions and their parameters, we need information on how those actions relate to the parameters, and what do they represent in the planning context.

For this, we propose to tag the domain file with information on how to generate the sentences. Thus, our methods require the input domains to be tagged with semantic information. While this introduces some manual work on the side of the domain expert, we believe this can also be useful to encourage commenting those domains, making it easier to understand the meaning of each action by the users of the domain. Therefore, we propose a commenting format to add the semantic information on the actions. Thus, we denote those tags as "semantic information tags" as they will help the domain readers to understand the semantics of the action without the need of digging into its conditions and effects. The tags describe the syntactic information from the actions and their parameters.

We propose a flexible approach to obtain the necessary information to verbalize the actions in the domain. Instead of writing all the template verbalization sentences, we tag each action and predicate with the *verb* that they represent along with its syntactic complements, and the *subject* of the sentence. These tags may include the parameters of the actions which will be replaced by their grounded value in the plan.

Our proposed format allows for the specification of alternatives to produce richer verbalizations (i.e., synonyms for verbs), which will be selected at random. For optional complements such as some prepositional clauses, we also have the option to flag them as required to avoid them being omitted based on the verbalization space parameters (as detailed in the following section). Alternative forms of the syntactical clause are separated by a forward slash (/), while prepositional clauses can be flagged as required by adding an exclamation mark (!) at the end. Phrasal verbs can be added by putting the particle in parentheses such that only the non-parenthesised part will be conjugated. For instance, the phrasal verb "look for" would be set up as `; verb = look (for)`. Figure 1 shows an example of a tagged action and predicate/fluent for both PDDL and RDDL in a robotics domain, including different verbal options.

These tags are then used to generate sentences for each action. The verb is conjugated to the appropriate tense by using the `mlconjug3` library (Diao 2020). Thus, our method is able to generate sentences in past, present, and future, which allows the planning system to update the verbalization of the plan while it is being executed.

## Task plan verbalizations

Following the definition from Rosenthal, Selvaraj, and Veloso (2016), we will define the *verbalization* of a task plan as the process that converts the plan into a natural language description. A natural language description of the plan may be easier to understand by a wider range of users, including non-experts in planning nor the domain. This understanding can then be key to increasing the trust of the users in the plan, as well as its transparency. User's acceptance can increase when the reasons for the system's actions are explained (Koo et al. 2015).

We propose a verbalization method that is domain-independent provided that the input domain has been tagged appropriately, as described above. For this, we use the ROS-Plan system (Cashmore et al. 2015) as the planning frame-

```
; Moves the robot from waypoint ?from to waypoint ?to
; verb = go / travel / move
; subject = ?v
; prep = from the ?from
; prep = to the ?to / towards the ?to !
(:durative-action goto_waypoint
  :parameters (?v - robot ?from ?to - waypoint)
```

```
; The robot ?r is at the waypoint ?wp
; verb = be
; subject = ?r
; prep = at the ?wp
(robot_at ?r - robot ?wp - waypoint)
```

(a) Example of PDDL action tagging.

(b) Example of PDDL predicate tagging.

```
// Moves the robot from one waypoint to another
// verb = go / travel / move
// subject = \1
// prep = from the \2
// prep = to the \3 / towards the \3 !
goto_waypoint(robot, waypoint, waypoint): { action-fluent, bool,
                                            default = false };
```

```
// The robot is at the waypoint
// verb = be
// prep = at \2
robot_at(robot, waypoint): { state-fluent, bool,
                             default = false };
```

(c) Example of RDDL action fluent tagging.

(d) Example of RDDL state fluent tagging.

Figure 1: Examples of semantic tags representing syntactic information of the planning actions and predicates. In RDDL, the parameter reference uses positional arguments such as $\backslash i$ for $i \in [1..n]$ for a fluent with $n$ parameters.

work. This allows us to have a planner-agnostic method, as well as to be able to work with both PDDL-based planners and RDDL-based planners (by using the probabilistic extension by Canal et al. (2019)). Note that, in the case of RDDL, there's one caveat which is that we are constrained to the subset of it supported by ROSPlan. Thus, causality information and goals (if present) may not be appropriately captured by ROSPlan, restricting the amount of verbalization that can be performed by our method. We support durative and non-durative actions (PDDL2.1) but not processes or events (PDDL+).

**Verbalization space**

Different users will have distinct preferences or needs when it comes to obtaining task plan descriptions. An expert user may need a detailed, step by step description of the plan to find incongruities or erroneous actions. A lay user, instead, may prefer to read a summarised version of the plan, know what was performed to achieve the main goals, or get a summary of the actions that were applied on a particular object.

In order to cope with these different verbalization use cases, we have extended the concept of *verbalization space* suggested by Rosenthal, Selvaraj, and Veloso (2016) to cover the narration task plans. The verbalization space specifies different variations of the descriptions of the plans to cover different user preferences. Our verbalization space for task plans includes four parameters: abstraction, locality, specificity, and explanation, as detailed below.

The combinations of the different parameters allow to generate various plan descriptions, from more detailed to more abstract and summarised, covering for a wide range of situations. This verbalization space for task plans should be general enough for most of the use-cases, but can easily be extended to deal with more parameters or combinations of them.

**Abstraction**   The abstraction parameter $a \in A$ represents the level of concretion used in the verbalization of the plans.

We consider four levels of abstraction:

- Level 1: The lowest level does not include any abstraction. This means that the verbalization will include numerical values such as real-world coordinates of objects or locations. It also includes the duration of the actions (if available), as well as all the action parameters. For this level, an extra file with the mapping between object instances and real-world data can be provided.

- Level 2: In this level, the parameter names are used instead of the available real-world values. It still verbalizes action durations and all the parameters, as well as intermediate values for compressed actions.

- Level 3: The duration of the actions is not verbalized, while all the parameters and intermediate values (such as via points) for compressed actions are kept.

- Level 4: In the more abstract level, only the essential parameters of the actions are verbalized, which are those needed for a grammatically correct sentence and the ones flagged as required. Intermediate values are also skipped.

**Locality**   The locality parameter $l \in L$ is used to narrow the scope of the verbalization, to perform it only based on points of interest of the user or a range of actions. We define three values for the locality:

- All the plan: Does not restrict the scope, and all the actions in the plan are verbalized.

- Range of actions: Restricts the scope to a subset of the actions of the plan. For instance, the verbalization would only take from the third action to the fifteenth one.

- Action or object: Limits the verbalization to those actions including a specific object instance as a parameter, or verbalizes all the actions with a given name.

**Specificity**   The specificity parameter $s \in S$ describes how specific the description of the plan should be regarding the level of detail. It includes three options:

- **General picture**: Provides a generic description of the main highlights of the plan. It focuses on the actions achieving the goals, and verbalizes those actions along with their justifications, provided that they are set so by the explanation parameter.

- **Summary**: The verbalization of the plan will compress actions when possible, generating a more compact representation of the plan. These compressions include shortcutting actions that act on intermediate objects (such as navigation actions through some via points), or join actions that are repeated with different objects or subjects. This is further detailed in the following section.

- **Detailed narrative**: Generates a detailed description of the plan without summarising nor compressing any action. Thus, all the actions will appear in the narration of the plan.

**Explanation**    The explanation parameter $e \in E$ specifies the level of justifications between actions that will be narrated. We have considered three kinds of verbalizable justifications: immediate justifications of actions, deferred justifications of actions, and goal-achieving explanations.

An action $a_j$ is an *immediate justification* of another action $a_i$ if $\forall k \in [i..j)$, there is a causal link between $a_k$ and $a_j$, where $i$, $j$, and $k$ are the indices in which the actions appear in the original plan. Thus, $a_j$ will be an immediate justification of all the actions in $[a_i..a_j)$, which are the actions that allow $a_j$ to happen.

A *deferred justification*, instead, happens when an action $a_i$ has a causal link with a non-consecutive action $a_j$. Therefore, we have a deferred justification when $\exists k \in [i..j)$ such that $a_k$ does not have a causal link with $a_j$.

*Goal-achieving explanations* make explicit the achievement of a goal, and show when an action was performed to complete a specific goal.

Following are the levels of explanation verbalizations:

- **Level 1**: No explanation is added to the plan, thus actions are verbalized sequentially in order of appearance.

- **Level 2**: Joins actions together when one action is an immediate justification of another action, and verbalizes them making the causality between the actions explicit.

- **Level 3**: Adds deferred justifications for actions that have a causal link with another action that appears later in the plan, but only if that action that is being justified achieves a goal. Deferred justifications to actions that act as an immediate justification are not verbalized.

- **Level 4**: The explanations of the goals that are achieved by the actions are added to the verbalization, along with the explanations from the lower levels.

- **Level 5**: Includes all the deferred justifications (for all the causal links of an action).

## Plan summarisation through action compression

It is often the case with some domains that the same action is sequentially repeated throughout the plan, with the in-between appearances of the action providing intermediate values that may not be very informative to the user.

(a) Compression with multiple objects. The resulting action symbolizes "r will grasp A, B, and C"

(b) Compression with multiple subject and the same parameters. The resulting action represents that "$r_1$ and $r_2$ will go from A to B"

(c) Compression with intermediate parameters. The resulting action means "r will go from A to D"

Figure 2: Examples of action compressions for plan summarisation

Examples of this include navigation actions for a robot, where it can only move between the waypoint it is at to another waypoint connected to it. Thus, to reach a certain position, it must traverse a set of these waypoints, generating many actions that reach intermediate positions. Similarly, a consecutive sequence of the same action applied to different objects can be summarised as the action applied to the set of objects.

We propose an action compression method to deal with these kinds of actions to generate shorter verbalizations of the plan. We only compress actions in the two aforementioned cases, and when there is only one free parameter (i.e., a grounded parameter whose value does not appear in both actions). Nonetheless, the method is easily extendable to handle more complex situations.

Given two consecutive appearances of an action in the plan, we compare their grounded parameters to generate a pattern that indicates whether each parameter had the same value in the two actions, or whether the same instance appeared in different parameter positions. We then perform the compression as follows:

- When all the parameters of the action but one have the same values in the same position, the resulting compression keeps those parameters and joins the free parameter in a list. Notice that this method will compress parameters that act as an object to the action and also those acting as a subject. Figs. 2a and 2b show examples of this.

- When the same grounded parameter appears in different positions in both actions, we consider that parameter to be an intermediate one. The resulting compression removes the intermediate parameter and joins both actions by keeping the rest of parameters. To do so, the space left by the intermediate parameter is filled by the grounded

values appearing at the same place in the other action. The intermediate parameters are kept to be used in the verbalization with abstraction levels 1 to 3. An example of this kind of compression can be found in Fig. 2c.

The compression method starts at the beginning of the plan and checks every pair of consecutive actions trying to compress them according to the above procedure. When two actions are compressed, the resulting action is compared with the next one, extending the compression to the subsequent actions in the plan. The compressed action duration is computed as the time overlap between the two actions.

## The PlanVerb algorithm

The plan is pre-processed and stored in an intermediate structure that will later allow the generation of the verbalized sentences. This structure is a script of the plan to be verbalized. Each element $s_l \in V$ in the script $V$ is a 4-tuple $s_l = \langle a_i, I_{a_i}, D_{a_i}, G_{a_i} \rangle$, where $a_i$ is an action, $I_{a_i}$ is a list of immediate justifications (actions $a_j$ that have a causal link *to* $a_i$), $D_{a_i}$ is a list of deferred justifications (actions $a_k$ that have a causal link *from* $a_i$), and $G_{a_i}$ a list of the goals achieved by $a_i$.

We first compute the action causality chains from the plan. For this, we use a graph-based representation of the plan, such as the one from Lima et al. (2020) that is integrated into ROSPlan. From the plan graph, we compute the causality chains for those actions achieving a goal by traversing the causal edges of the graph from these goal-achieving actions backwards.

Algorithm 1 shows the pseudocode of the PlanVerb algorithm. To start, the actions in the plan are compressed using the COMPUTEPLANCOMPRESSIONS method (line 5), as described in the section above. The compression method splits the plan into one plan per each subject performing an action in the plan. This enhances the number of action compressions, as only actions appearing consecutively in the plan are compressed. Thus, actions are considered in a per-subject manner instead of in the whole plan.

Then, the causality chains are used to generate a full plan script integrating all the information (immediate justifications, deferred, and goals) for every action in the plan. We call this script the causality script, and it is generated in COMPUTECAUSALITYSCRIPT (line 2). The justifications are also considered on a per-subject basis, so immediate justifications may not be consecutive in the general plan, but be in the plan split by subject.

The causality script is then iterated, and the verbalization space parameters are applied, generating a verbalization script that includes the actions that will be finally verbalized. The actions and action justifications are filtered based on the verbalization space parameters (lines 12–15). Actions acting as an immediate justification of another action are skipped and not included in the script, given that they will be verbalized along with the action they support. Actions appearing in a deferred justification are not skipped. Instead, they are verbalized both as a (later) consequence of the causing action when it is verbalized, and as an action with its own justifications when it appears later in the plan.

---

**Algorithm 1:** The PlanVerb algorithm

**Input:** Plan $\pi$; Causality chains $C$; Semantic tags $T$
      Verbalization space $(a, l, s, e) \in (A, L, S, E)$
**Output:** Verbalization $v$

1   $GA := \text{GETGOALACHIEVINGACTIONS}(C)$
2   $CS := \text{COMPUTECAUSALITYSCRIPT}(\pi, C)$
3   $PC := []$
4   **if** $s == Summary$ **then**
5      $PC := \text{COMPUTEPLANCOMPRESSIONS}(\pi, GA)$
6   **else if** $s == General\ Picture$ **then**
7      $CS := \text{GETGOALACHIEVINGSCRIPTS}(GA, CS)$
8   $v := []$
9   **foreach** $c \in CS$ **do**
10      **if** $\text{NOTINLOCALITY}(c, l)$ **then** skip $c$
11      **else** // Filter the scripts according to e
12          $c.I := \text{FILTERIMMEDIATEJUSTIFICATIONS}(c.I, e)$
13          $c.D := \text{FILTERDEFERREDJUSTIFICATIONS}(c.D, e)$
14          $c.G := \text{FILTERGOALS}(c.G, e)$
15          $v.\text{add}(\text{GENERATESENTENCE}(c, a, T, \pi, PC))$
16   **return** $v$

---

To avoid over cluttering the sentences, deferred justifications are skipped when they justify an action that has been skipped (i.e., it acts as an immediate justification to another action), or when they appear in a sentence where a goal is verbalized and the explanation level is lower than 5 (thus, goals take precedence).

**Sentence generation**   Each action in the script is verbalized in line 15 of Algorithm 1. The GENERATESENTENCE method checks whether there are immediate, deferred justifications, or goals in the script, verbalizes each of them and joins them with pre-defined sentence linkers. The selected linker is chosen at random, and the actions are verbalized to the appropriate tense based on the structure of the linker and the tense of the main action in the script.

The script may be tensed in future, past, or present depending on the execution point of the plan. The justifications and deferred justifications are tensed accordingly, with the verbs conjugated using the `mlconjug3` library. All the actions and predicates (goals) are verbalized similarly, taking the form of `subject + verb + indirect-object + direct-object + prepositional clauses`, using only the available parts of the sentence based on the semantic tags of the action and the abstraction parameter.

The sentence generation process also checks whether the actions in the script are compressed and uses the compressed version, adding the intermediate values as via points and the action duration when specified by the level of abstraction.

## Evaluation

We have evaluated the proposed plan verbalization method and spaces. First, we provide some examples of automatically verbalized actions. Then, we analyse the impact of the

verbalization space parameters. Finally, we comment on the results of an online survey regarding the verbalization.

For all the examples and results below, we have user ROS-Plan with the POPF planner (Coles et al. 2010) for PDDL domains, and the PROST planner (Keller and Eyerich 2012) for RDDL domains[1] .

## Examples of verbalized actions

Here we will present some verbalized actions produced by our algorithm. In this section, we will use a robotics domain where mobile robots work in an office performing navigation, pick and place, and handover tasks. In the exemplified plans, two robots are acting in the environment, where one of the robots is the narrator, speaking in the first person, and the other one is called "Tomo".

In the following examples, black sentences refer to the main action, blue sentences to immediate justifications, green sentences to deferred justifications, and red sentences to goals. We have added sentences in different tenses to show the ability of the method to generate sentences at different points of execution.

**Example 1: Abstraction**   We start with an action appearing early in the plan where Tomo locates the manager. This action enables the actions of "request person" and "give object", being the latter achieved by the other robot at the last part of the plan. The sentence verbalized with $(a, l, s, e) =$ (Level 3, All plan, Summary, Level 4), action durations are not included, and all the parameters are verbalized.

> Tomo will locate the manager, which will allow me to **later** request the manager at the kitchen corridor and me to hand post2 to the manager at the kitchen corridor.

If verbalized with abstraction level 4, the resulting sentence ignores the location prepositional clause:

> Tomo is going to locate the manager, which will allow me to **later** ask the manager and me to deliver post2 to the manager.

**Example 2: Specificity**   Here we'll show how actions get compressed in the plan. With verbalization space parameters $(a, l, s, e) =$ (Level 3, All plan, Summary, Level 4), intermediate actions are compressed as follows:

> Tomo will travel from the main office desk towards the kitchen chair (via main office doorway and meeting room hallway) to move from the kitchen chair to the entrance to achieve the goal of Tomo being at the entrance.

When using the Detailed Narrative value instead, the sentence is the following (this time in past tense). Notice that

---

[1]The code, domains, and the complete set of verbalized plans with all the combinations of verbalization space parameters can be found in https://github.com/gerardcanal/task_plan_verbalization

as there is no compression employed, actions from different subjects are interleaved:

> Tomo went from the main office desk towards the main office doorway to then go from the main office doorway to the meeting room hallway. I traveled from the kitchen corridor towards the kitchen shelf to reach the goal of me being at the kitchen shelf. Tomo went from the meeting room hallway to the kitchen chair so Tomo could travel from the kitchen chair to the entrance to fulfill the goal of Tomo being at the entrance.

In the following example from the IPC 2002 Rovers domain (Long and Fox 2003), subjects are compressed. With verbalization space parameters $(a, l, s, e) =$ (Level 2, All plan, Detailed Narrative, Level 1), the verbalization is:

> Rover3 will travel from waypoint7 towards waypoint0 (taking 5 seconds). Rover2 is going to travel from waypoint7 towards waypoint0 (taking 5 seconds).

When compressed with the Summary specificity parameter and set to present tense, it becomes:

> Rover3 and Rover2 are traveling from waypoint7 towards waypoint0 (taking 5 seconds).

**Example 3: Immediate and deferred explanations**   Finally, we show an example of sentence verbalized with both immediate and deferred justifications. For this one, the verbalizations parameters are $(a, l, s, e) =$ (Level 4, All plan, Summary, Level 4):

> Tomo is going to move to the office entrance 1, which will allow Tomo to grasp the post1 so Tomo can **later** leave the post1 at the main office desk.

## Effect of the verbalization space parameters

We have validated the effect of the different verbalization space parameters with a set of test domains. Those include the office robot domain (4 instances), the IPC 2002 Rovers domain (Long and Fox 2003) (19 instances), the IPC 2008 CrewPlanning domain (Barreiro, Jones, and Schaffer 2009) (30 instances) for PDDL. For RDDL, we have used the IPPC 2014 triangle tireworld (Little, Thiebaux et al. 2007), the print fetching domain from (Canal et al. 2019), and three interactive assistive robotics domains (Canal 2020), where the tasks involve a robotic assistant feeding and dressing a human.

We have computed a plan for all the domains and instances and verbalized it with all the combination of parameters. Fig. 3 shows the average number of words for all the verbalized plans in different combinations of parameters.

Regarding the abstraction parameter (Figs. 3a and 3b), the figures show that the higher the level of abstraction, the

fewer the number of verbalized words. Note that for abstraction level 1, in this experiment we have sampled random real coordinates from a 2D space to represent the locations appearing in the problem instances.

For the explanation parameter (Figs. 3b and 3c), the number of words increases with the level of explanation, as the text becomes more verbose. Level 2 has a slight increase, as the same number of actions are verbalized but linked together. The deferred justifications added by level 3 increase more the number of words, surpassed by level 4 with the verbalized goals. Lastly, the inclusion of all the deferred justifications in level 5 generates the largest verbalizations.

The specificity parameter (Figs. 3a and 3c) also has a clear effect on the number of words. The General Picture is the most summarised one, including only some actions. The Summary level includes all the actions but compresses some of them, thus producing shorter narrations than with the Detailed Narrative parameter.

As can be seen in the different plots of Fig. 3, the verbalization space parameters are consistent with the generated plan narratives.

## Online user survey

We have conducted an online survey to assess both the usefulness of the provided explanations and the understandability of the generated sentences. The survey was answered by 42 people in two groups. Two verbalizations were shown to each of the users. The first one, $v_1$ is a step-by-step plan of two robots, Tomo and Asro, performing the tasks of the office domain. The narration for $v_1$ was generated with parameters $(a, l, s, e)$ = (Level 3, All plan, Detailed Narrative, Level 1). The other one, $v_2$ is a summarised version of the same plan including explanations, generated with the parameters $(a, l, s, e)$ = (Level 3, All plan, Summary, Level 4). One group would see first the step-by-step plan $v_1$ and then the summarised one $v_2$; the other would see them in the opposite order. The background of the users ranged from robotics, computer science, AI planning, and unrelated disciplines (non-technical). The 57% of the users not familiar with task planning (lay users), and the 28% had occasionally seen or used a task planner before (non-experts). Four users were considered expert. We kept them in the analysis because, while our focus is on non-expert users, we wanted to see if there were notable differences in views or comments from them. We did not find difference in performance, while we got meaningful opinions from them. All of the users were fluent in English. The survey involved multiple-choice questions on their opinions on why they thought some specific actions were happening in the plan based on the goals of the robots. The multiple-choice questions were followed by 5-point and 7-point Likert-scale questions regarding their agreement with different statements on the understandability of the plans. Finally, some open-ended questions concluded the survey. In the statistical tests used to analyse the results of the survey, which we will now discuss, we have used a confidence level of 95%.

An F-test showed there were not significant differences between the two groups, for which the following results will aggregate the answers regarding $v_1$ and $v_2$ for both groups.

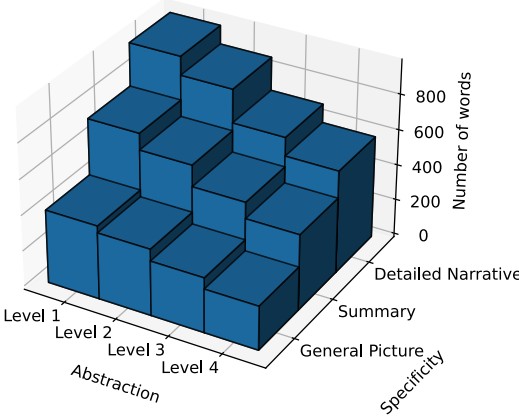

(a) Average number of words in the verbalizations for the abstraction and specificity parameters

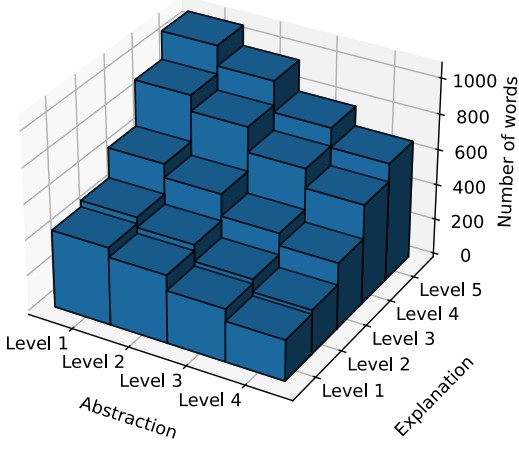

(b) Average number of words in the verbalizations for the abstraction and explanation parameters

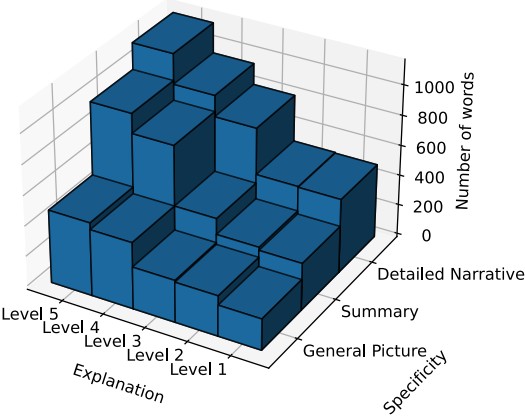

(c) Average number of words in the verbalizations for the explanation and specificity parameters

Figure 3: Effect of the verbalization space parameters in the average number of generated words

The answers to the multiple-choice questions were given one point for a correct value, and half a point for partially correct answers (for instance, when the answer involved two reasons but only one of them was selected). Our results clearly show that $v_2$, which included justifications, helped the users to better answer the questions. More than 80% of the users were able to answer correctly, with the highest question being 97.62%. In contrast, for the step-by-step description $v_1$ (without justifications) only half of the users gave a correct answer, with the maximum for a single question being 61.90%. For each question, between the 20% and 40% of the users stated they did not know the answer for $v_1$, while this percentage was at most 2.38% for $v_2$. We have assessed the significance of these results with a $\chi^2$ test.

Regarding the Likert questions, users were more confident in their responses for $v_2$ ($\tilde{x} = 5.67^2$ out of 7) than for $v_1$ ($\tilde{x} = 4.11$ out of 7). Regarding how easy it was to answer, the means were $\tilde{x} = 4.02$ out of 5 for $v_2$ and $\tilde{x} = 2.45$ for $v_1$. Those results indicate that users found explanations to be helpful to answer the questions, clearly shown by the answers to how quickly they could find the reasons behind each questioned action. For this, the answers were $\tilde{x} = 4.12$ out of 5 for $v_2$ and $\tilde{x} = 2.30$ for $v_1$. Users also found the descriptions in $v_2$ easier to read and understand. Similarly, they reported higher satisfaction with the plan description for $v_2$, which included explanations. A t-test found statistical significance for all these answers.

When asked about the grammatical soundness of the generated sentences, there were no significant differences between both verbalizations. For $v_1$, $\tilde{x} = 4.19$ out of 5, while $v_2$ got a $\tilde{x} = 4.17$. Thus, both representations generated by PlanVerb were found to be correct and readable regardless of the parametrised level of explanation.

When asked by whether the description of the plan made it easier to answer the questions, most of the users agreed for $v_2$ (with justifications) against $v_1$ (without justifications). Users that got $v_2$ last agreed it helped more than $v_1$ with a mean of $\tilde{x} = 4.85$ out of 5, while those seeing $v_1$ last gave a mean score of $\tilde{x} = 2.14$ regarding $v_1$ helping more than $v_2$. This supports the claim that the justifications help users understand the reasons for the actions. A t-test also showed significance for these results.

When asked about improvements, some users pointed to elements that can already be solved by the different combination of verbalization space parameters, such as showing only actions achieving goals. Some users believed there was too much information, with excessive granularity in $v_1$, while others mentioned $v_1$ was missing information while $v_2$ wasn't. This appears in different answers and suggestions such as adding temporal information (which we can do with different parameters). Therefore, users' answers clearly support the need for different parametrizations, given that users will have their preferences over the best verbalization. A few users mentioned that one plan for each robot would be easier to understand, which supports the idea of joining plans by subject and summarizing them separately as we have proposed. Finally, many users suggested adding different kinds

of visualisation along with the verbalization, with different ideas that we leave for future work.

The conducted survey demonstrates how PlanVerb can generate narrations of task plans that make sense to users and are grammatically sound. The users' answers further support the need for different kinds of verbalizations, which can be achieved with the verbalization space parameters we have proposed.

## Conclusions

In this paper, we have presented PlanVerb as a domain-independent method to automatically verbalize task plans. We have proposed a semantic tagging for PDDL/RDDL actions and predicates that provide the necessary information for PlanVerb to generate natural language sentences. Then, by using causality information between actions, we are able to generate sentences that make this causality explicit, both for immediate justifications of actions appearing consecutively in the plan and for deferred justifications of actions that appear at a later stage. The narrated plans can also be summarised by compressing related actions. We have further extended the concept of verbalization space introduced by (Rosenthal, Selvaraj, and Veloso 2016) used to narrate robot navigation experiences to cover task plans. We have added a new parameter of explanation, which decides the number of justifications that are verbalized, as well as the filtering by object or action introduced in the locality parameter.

We have shown examples of verbalized sentences using our method and evaluated the effect of the verbalization space parameters in different domains. An extended set of examples can be found in the repository[3]. Finally, we have conducted an online survey where users were shown examples of verbalized texts. All the users were able to read the texts and confirmed the sentences were grammatically sound. Moreover, the justifications helped them to understand the plan, easing the process of answering questions about the plan, supporting the hypothesis that verbalizing causal chains fosters plan understanding. We believe this is a good step towards making task plans more understandable by lay users and users unfamiliar with the domains. However, users' answers also pointed to the need for better Explainable Planning (XAIP) methods, able to explain the underlying reasons for some of the actions beyond making action causality explicit.

Although we can successfully verbalize plans that are understandable by users, there are still some improvements that can be done as future work. First, using some natural language processing techniques to improve the generated sentences. Pronominalisation could help to make sentences more natural by avoiding subject repetition, as well as pluralization of nouns when needed (i.e., after some action compressions). Finally, the addition of preconditions and effects could be beneficial to the verbalization process, along with an improved justification selection mechanism. This could be accompanied by plan visualization techniques, that clarify the different steps involved in the plan.

---

[2]Where $\tilde{x}$ represents the arithmetic mean.

[3]Examples can be found at https://github.com/gerardcanal/task_plan_verbalization/tree/with_examples/verbalized_examples

## Acknowledgments

This work has been supported by the EPSRC grant THuMP (EP/R033722/1). The authors would like to thank Mr. Sekou Diao for his insights and help with the use of `mlconjug3`, and Mr. Ionut Moraru, Mr. Alexander Ortiz de Guinea, Dr. Michael Cashmore, Dr. Rita Borgo, and Dr. Xavier Ferrer for fruitful discussions.

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
