# OpenReview forum: "Task Plan verbalizations with causal justifications"
_icaps-conference.org/ICAPS/2021/Workshop/XAIP — XAIP 2021_

### Official Review · AnonReviewer1 · 2021-07-04
**Interesting approach for plan verbalization**

**Rating:** 7
**Confidence:** 4

**Review:**

## Paper Summary

The paper proposes a domain-independent plan verbalization approach called PlanVerb. The proposed approach is capable of generating natural language descriptions of plans, including explanations of causality between actions of a plan. The authors extend the verbalization space of previous work, and apply it for generating different types of plan descriptions according to user's preferences. PlanVerb can deal with both PDDL and RDDL domains. An evaluation with 42 participants shows that PlanVerb provides understandable plan verbalizations.

## Review

The paper is overall very well written, organized, and easy to follow. I think that the authors could try to formalize the Verbalization Space in a more formal way and try to improve the explanation about Algorithm 1 (perhaps add a workflow chart), but apart from that, I really like the paper and the idea of plan verbalization.

Please, consider the following minor points and comments:

### Minor Points and Comments:

1. The authors could have explained the main differences between PDDL and RDDL. Some readers are not familiar with RDDL and its formalism. Please, consider adding a comment about the differences and similarities between PDDL and RDDL.

2. The authors say that PlanVerb does not need pre-writing sentence templates, only tagging the syntactic elements of the actions in the domain model. However, the semantic domain information tagging actually defines some sort of template in the action description (commenting format) to generate sentences. Please, clarify this.

3. As I mentioned before, it would be interesting to see a formal definition of the Verbalization Space. Some symbols are introduced in The PlanVerb Algorithm section without much explanation. Please, try to formalize the Verbalization Space before introducing Algorithm 1.

4. In the PlanVerb Algorithm section, the explanation and notation in the first paragraph seem a bit out of context with respect to Algorithm 1. The notation introduced in that paragraph is not used in Algorithm 1 or in any other part of the paper. Please, check whether such notation is really needed in that section, and if not, please consider removing it.

5. Algorithm 1 (The PlanVerb Algorithm) is somehow useful to understand the plan verbalization process, but it would be good to see it as a workflow chart, illustrating the verbalization process and etc.

6. I have some questions and comments about your evaluation, as follows.
	- 6.1) You say: "... we have used P = 0.95 as the confidence level.", what does it mean?
	- 6.2) Please, clarify what \tilde{x} means.
	- 6.3) Please, explain the tests you used in your evaluation. You mention t-test and etc, but you don't say what they actually do.

7. I could not find the supplementary material in the paper submission.

8. Typos and writing issues:
	- a quick idea of what does each action represent. -> a quick idea of what each action represents.
	- Includes of all the deferred justifications -> Includes all the deferred justifications;
	- different combination of parameters ->  different combinations of parameters;
	- we wanted so see -> we wanted to see;
	- Please, revise this sentence: "For each question, there was between a 20% and a 40% ...";
	- with the highest question being the 97.62% -> with the highest question being 97.62%;
	- More than the 80% -> More than 80%.

## Final Remarks
To conclude, in my opinion, I think that this paper is relevant to the workshop and should be accepted.
Please, try to address the minor points and comments that I pointed out above in the next revision/version of the paper.

---

> ### Author Response · Authors · 2021-07-29
> **We appreciate the comments of this reviewer**
>
> We would like to thank this reviewer for all the comments and useful suggestions. Following we'll answer the different questions they posted:
>
> 1. We have added citations that were missing on PDDL and RDDL. RDDL is the Relational Dynamic influence Diagram Language. It allows for stochasticity and partial observability and has been used in many IPPC editions.
>
> 2. We do need the tags, and the tags indeed define a kind of template. However, we found those templates to be more flexible than writing the full sentence, as this provides the basic building blocks to define different kinds of sentences, resulting in a more varied verbalization. We also see the commenting as a nice side-effect of the verbalization, as the tags can provide insight on what does each parameter of the action represents, and what is the action doing, similar to how docstrings are used in programming languages.
>
> 3 and 4. The notation introduced is used in the algorithm, in lines 12-15 and that is why it was introduced. We did find some typos in that part of the algorithm though, and they have been fixed.
>
> 5. We appreciate the suggestion and we'll try to prepare something that clarifies the process.
>
> 6.1 - This means the confidence interval for our statistical tests is 95% (which is the value usually employed for these kinds of tests). We have rewritten the text to make it clearer.
> 6.2 - \tilde{x} represents the arithmetic mean. We have added this to the manuscript.
> 6.3 - The t-test is an inferential statistical test used to determine if there is a significant difference between the means of two groups. It is a hypothesis test usually used to test the significance of obtained results. The Chi² test checks if there's a statistically significant difference between frequencies.
>
> 7. We apologize as we could not attach the supplementary material to Openreview. We have instead updated the manuscript with a link to an extended set of examples.
>
> 8. We appreciate the notes, and we have fixed the issues in the updated manuscript.

---

### Official Review · AnonReviewer2 · 2021-07-06
**A Useful extension of plan verbalization to Task Plans.**

**Rating:** 7
**Confidence:** 4

**Review:**

## Goal of the paper

The paper addresses an important problem of verbalizing plans generated by automated planners such that it is more accessible to non-expert users.

## Significance to the area

It is indeed relevant to the XAIP community as it contributes to making plans comprehensible to experts as well as lay users in natural language.

## Summary

The work proposes the PlanVerb framework which allows for verbalizing plan outputs from automated planners in natural language, while permitting the end user to exert control over various dimensions of verbalization like abstraction, locality, specificity and justifications. The paper, in my opinion, has the following contributions :

1.  Extension of Verbalization space (Rosenthal et al 2016) to Task Space. They support more domains (other than navigation) across PDDL and RDDL. (Domain - Independent method).

2.  Addition of another dimension : Explanations : They introduce immediate, deferred justifications, and goal explanations, to incorporate causal links between actions in the verbalization.

3.  Plan Compression : Under "Summary" value of Specificity dimension, plan compressions essentially allows for shorter verbalization by joining descriptions of actions and or subjects. (For example instead of saying Robot R1 went to A. Robot R2 went to A. Plan compression may produce : Robot R1 and R2 went to A.)

The authors also make use of mlconjug3 (Diao 2020) to assign appropriate tense to the verbs.

The key ingredient required to make the verbalization possible is semantic annotation of each action to specify subject, verb, preposition etc. Further synonyms can also be provided to the given verbs along with flags which make those mandatory to be included in the verbalization. This is assumed to be manually input to the system.

Finally, the authors have shown a well formed user study to substantiate that Plan Compression and addition of explanatory utterances and justifications is very helpful. The results of study seems sound and helps in justifying their claims.

## Comments:

Although I'm not an expert in the area, but the authors have have done a good job at covering the related works. However, I would have liked a longer discussion on comparing to works using casual links for verbalization/explanations, specially because a large contribution (and the corresponding study) covers immediate and deferred justifications (which are similar to distal explanations, which again is cited but has not been discussed at length). Another interesting discussion could have been in comparing verbalization with "drill down" operations in Big Data which essentially allows business facing users to expand on and abstract away along various dimensions.

The discussions on verbalization space and individual components of each dimension like levels in abstraction etc are well explained, however they lack accompanying examples (There are a few at the end, but a long running example could have helped understand how each level can affect the verbalization - Question on this at the end).

The PlanVerb algorithm (Algorithm 1) helps understand the flow of how final action selection will be made, which would be further verbalized. The Proposed Work is clearly stated and was mostly easy to understand.

Although I'm not familiar with all the recent works, in my opinion, the extension that this work proposes is very useful since it allows verbalization beyond robot navigation. They have extended all the dimensions - Abstraction, Specificity and Locality from the original work of Rosenthal et al 2016 to make it domain independent. They proposed the additional dimension of Explanations which indeed makes a lot of sense and they have found it to be useful in their studies as well. Finally they have put in efforts to make the final sentence to be as "natural" by leveraging mlconjug3.

## Questions/Concerns
Overall I liked the paper & and its contributions, however I do have the following questions/concerns to the authors & hopefully answers to these will help future readers/authors.

1.  Although as a preliminary work it makes sense to assume manual work in obtaining semantic tags, in the long term, I believe the domain designers should be made separate from experts providing the semantic tags. This is because the tags are being used as it is and delivered to the users by filling up existing templates. Hence, many times terminologies which are easy to understand for domain experts might not be the best choice of terms for lay users.

2.  Authors mention for Level 4 under abstraction dimension of the Verbalization Space that "only the essential parameters of the actions are verbalized". It was unclear to me what these essential parameters are and could be chosen.

3.  Maybe its straightforward, but it seems the description on "goal-achieving explanations" is missing from Explanation subsection under Verbalization Space. Similarly it is used for Level 4 for Explanations, but I had to assume its meaning from the examples later on.

4.  Authors say "Compression method starts at the beginning of the plan and goes through every pair of actions", it should be consecutive actions.

5.  In algorithm 1, function role of FILTERGOALS is unclear to me.

6.  Example 1 : Abstraction uses the following setting (a,l,s,e) = (L3, All Plan, Summary, L4). Since Explanation is set to Level 4, goals should also have been conveyed in the verbalization. But I'm confused if conveying the goals is optional or mandatory under Level 4. If it is optional, how is it decided.

7.  Again, for Level 3 : Abstraction, (take Example 1 which uses L3 abstraction & presents another example with L4 abstraction), the authors say that example with L4 ignores location clause. On what basis did PlanVerb decide to keep details like ask the manager but drop "at the kitchen door", that is, how did it make sure that person of interest is kept while location is secondary. If infact there is a precedence for selection of the parameters across abstraction levels, doesn't this become domain dependent?

8.  Authors seem satisfied with the support for per-subject plan verbalization from their user study (".. supports the idea of joining plans by subject..") but it seems that the choice of which one should be better is completely dependent on the kind of domain. If the interactions between subjects is very often then it makes sense that per-subject verbalization is ignored, whereas if the actions are mostly independent like robot navigation, then yes, per-subject verbalization could be useful.

9.  Under PlanVerb Algorithm section the authors say that "the justifications are also considered on a per-subject basis". This means only the actions for each subject are compressed & justifications for each subject is computed. Maybe I'm wrong, but this should mean that justifications across subjects should have been ignored and never appear in the verbalizations. However, Example 1 is "Tomo will locate the manager, which will allow me ..." where the causal link between two subjects is part of the verbalization. This shouldn't be possible unless I'm missing something.

10. Maybe beyond the scope of this work, but it seems the verbalizations are abstracting away the order in which actions should take place across different subjects. Example 2 : Detailed Narrative specificity : takes about Rover3 going to waypoint 7 and rover 2 going to waypoint 7. What if the plan requires that the order is strictly rover 3 then rover 2. Would that still be captured by PlanVerb?

I hope my comments are useful.

---

> ### Author Response · Authors · 2021-07-29
> **Many thanks for the detailed review!**
>
> First of all, thanks to the reviewer for the nice and detailed review. Following we'll answer the different questions and concerns posed by them:
>
> 1. The idea with the semantic tags is also to have them as a user-friendly comment that explains what is the action doing (with many domains not being too user-friendly, this could help users of the domain understand the semantics of the actions). While domain experts have the potential of filling those tags, we do not intend to restrict the definition of the tags to a domain expert. For the verbalization algorithm, we do not mind who provides the tags as long as the input domain is tagged.
>
> 2. We refer to the elements that are needed for the sentence to make sense. For instance, direct objects would not be skipped as most of the verbs representing actions that have a direct object are transitive. Some prepositional clauses, in contrast, may be omitted and the sentence still makes sense. Thus, all prepositional clauses not flagged as required (using the ! flag in the semantic tag) are not verbalized.
>
> 3. We didn't define this element which is important. We have now added the definition under the Explanation subsection.
>
> 4. We have fixed this.
>
> 5. The filter functions remove elements from the scripts based on the Explanation parameter $e$. In the case of the FILTERGOALS, it will remove them if e < 4.  We have added a comment on the algorithm we hope will make this clearer.
>
> 6. That's a good catch. In Level 4, goals are indeed verbalized but only for actions that achieve a goal. For those actions that don't, the verbalization is the same as for level 3. In the provided examples, the actions were not achieving any goal, therefore goals were not verbalized.
>
> 7. This relates to the answer to question 2. Direct objects and indirect objects are always kept as they are required for the correct generation of the sentence. Prepositional clauses are omitted unless they are flagged as required using the "!" symbol on the semantic tag. This means that a prepositional clause is needed for the verbalization to make sense or be understood. In the example of moving a robot from A to B, saying where the robot goes it's required (otherwise the sentence "Tomo is going" would not make sense). However, saying the origin waypoint may be optional ("Tomo is going to the kitchen" makes sense, but we can also give more information saying "Tomo is going from the living room to the kitchen"). This is achieved by using the "!" flag as described in section "Semantic domain information tagging".
>
> 8. The per-subject plan separation was introduced to better deal with immediate justifications and compressions, which depend on consecutive actions. The per-subject separation does not influence the causal links between actions from different subjects, as shown in the examples. So far, this has worked out in all the tested domains. Nonetheless, it can easily be disabled for domains where this may be an issue.
>
> 9. This refers mainly to immediate justifications. As we consider the position of actions in the plan to make this distinction between deferred and immediate justifications, having an action of a different subject between two consecutive actions of the same subject may break an immediate justification. Thus, we consider actions to be consecutive or not on the per-subject plan instead of in the general plan. Causal links between actions from different subjects are kept.
>
> 10. Yes, in this case, this would be captured by PlanVerb. The actions in the plan are sorted before being verbalised by the time the action starts. In the case of sentences with immediate justifications, those are verbalized first if they appear first in the plan. While it may happen that some action ends verbalized earlier or later than in the original plan, at the end of the day we're writing text that must be written in a sequential manner. A way to mitigate this in cases where needed is to use the abstraction parameter to specify the start times of actions so that if an action gets misplaced this is explicit. In domains where this may be critical, other techniques could be more appropriate such as visualization of the plan, as some users already suggested.

---

### Meta-Review · Area_Chairs · 2021-07-07

**Recommendation:** Accept
**Confidence:** 4

**Metareview:**

The paper looks at the problem-generating domain-independent plan verbalizations and builds on previous efforts in this direction in the domain robot path planning. The method also provides the user a certain level of control in the kind of verbalizations that can be generated through updating various parameters (again similar to earlier works). Both reviewers seem to agree that the paper is well suited to the workshop, so I would recommend accepting the paper.

In terms of comments, the main concerns raised by the reviewers seem to be about the clarity of the paper, with both reviewers pointing at various information missing from the text, including information about the algorithms, notational issues, and the need to include more intuitive descriptions of statistical results and tests. The authors have also pointed out issues, like how the precedence between details to be abstracted are identified and how separating descriptions per agent would affect cases where there are close interactions between the agents.

I would definitely recommend the authors to not only address these comments in future drafts of the paper, but also to respond in the openreview.

---

### Decision · Program_Chairs · 2021-07-08

Accept